# Feasibility and acceptability of using the BabySaver resuscitation platform and NeoBeat together for neonatal resuscitation in a low-resource setting: A pre-post implementation study

Milton W. Musaba[1,2,3], Ritah Nantale[2,4]*, David Mukunya[2,3,4,5], Julius N. Wandabwa[1,3], Kathy Burgoine[3,6], Nicolas J. Pejovic[7], Benjamin A. Kamala[8], Agnes Napyo[2,3,9], Kenneth Mugabe[1,2,3], Brendah Nambozo[2,3,4], Faith Oguttu[2,3,4], John Stephen Obbo[10], Thorkild Tylleskar[3,11], Andrew D. Weeks[3,12]

**1** Busitema University Faculty of Health Sciences, Department of Obstetrics and Gynaecology, Mbale, Uganda, **2** Accelerating Innovations in Maternal, Adolescent, Reproductive and Child Health (AiMARCH), Mbale, Uganda, **3** Sanyu Africa Research Institute (SAfRI), Mbale Uganda, **4** Busitema University Faculty of Health Sciences, Department of Community and Public Health, Mbale, Uganda, **5** Department of Research, Nikao Medical Center, Kampala, Uganda, **6** Department of Pediatrics, Mbale Regional Referral Hospital, Mbale, Uganda, **7** Department of Global Public Health, Karolinska Institutet, Stockholm, Sweden, **8** Department of Obstetrics and Gynecology, Muhimbili National Hospital, Dar es Salaam, Tanzania, **9** Kabale University School of Medicine, Department of Nursing, Kabale, Uganda, **10** Department of Internal Medicine, Mbale Regional Referral Hospital, Mbale, Uganda, **11** Centre for International Health, University of Bergen Faculty of Medicine and Dentistry, Bergen, Norway, **12** Department of Women's and Children's Health, University of Liverpool, Liverpool Women's Hospital, Liverpool, United Kingdom

* ritahclaire24@gmail.com

## Abstract

### Background

BabySaver and NeoBeat devices have the potential to enable bedside neonatal resuscitation, with an intact cord in the presence of the mother. We assessed the feasibility and acceptability of using them together for neonatal resuscitation in a low-resource setting.

### Methods

This was a mixed methods study conducted over a period of 11 months at Mbale Hospital in Uganda. We enrolled 150 mother-infant dyads into a pre-post study. During the pre-implementation phase, neonatal resuscitation was conducted based on the existing standard of care whilst in the post-implementation phase we evaluated the BabySaver and NeoBeat. Our primary outcome was the proportion of babies resuscitated at the bedside with an intact cord. Using in-depth interviews and an inductive thematic analysis approach, we also explored experiences of health workers and mothers with use of the BabySaver and NeoBeat.

**Data availability statement:** All relevant data are within the paper and its Supporting information files.

**Funding:** This study was supported through the Saving Lives at Birth program in Low and Middle Income Countries under the Laerdal Foundation (#REF:2022-0264) The funding sources had no role in the study design, data collection, data analysis, data interpretation, the manuscript's writing, or the decision to submit the manuscript for publication. The content is solely the responsibility of the authors and does not necessarily represent the official views of the funding sources.

**Competing interests:** ADW was one of the co-inventors of the BabySaver tray. The intellectual property is held by his employer, the University of Liverpool, but the rights for Africa were sold to the Sanyu Africa Research Institute for £1 in 2019 so that they could take forward its development and distribution in Africa. ADW holds no personal rights to the IP, but there is a royalty sharing scheme at UoL whereby any future royalties would be shared with him. This does not alter our adherence to PLOS ONE policies on sharing data and materials. There are no patents, products in development or marketed products associated with this research to declare.

## Results

Bedside resuscitation increased significantly in the post-implementation period (9.3% versus 45.3%, $p < 0.001$ while early cord clamping decreased (26.7% versus 12.0%, $p = 0.042$). The median time to successful resuscitation was shorter post-implementation (5 versus 8 minutes, $p < 0.001$). Infants in the post-implementation phase had higher axillary temperatures at birth and at 0-, 10-, 20-, and 30-minutes post-resuscitation. Neonatal morbidity was lower: APGAR score <7 at 5 minutes (aPR: 0.36; 95%CI: 0.26–0.50), transfer to postnatal ward with mother (aPR: 9.27; 95%CI: 2.23–38.48), transfer to neonatal unit (aPR: 0.66; 95%CI: 0.56–0.78). Health workers found the devices easy to use, and bedside resuscitation reassured mothers, fostering trust and satisfaction. Barriers included misconceptions about delayed cord clamping, hypothermia concerns, cross-infection risks, and difficult use in theatre.

## Conclusion

The BabySaver and NeoBeat improved bedside neonatal resuscitation and reduced morbidity. Bedside resuscitation was also acceptable to the health workers and mothers. Scaling up should address misconceptions about delayed cord clamping and optimize usability in theatre settings where many asphyxiated infants are delivered.

## Background

Sub-Saharan Africa accounts for 42% of the 2.4 million neonatal deaths that occur globally each year [1–4]. It is estimated that 30% of these deaths are due to perinatal asphyxia, which is often related to intrapartum complications [4,5]. Uganda has a high neonatal mortality rate of 22 deaths per 1,000 live births and about 28% of these neonatal deaths are due to perinatal asphyxia [6,7]. Timely and effective neonatal resuscitation can prevent many of these deaths. However, efforts are often hindered by the lack of training and suitable or functional resuscitation equipment [8].

Neonatal resuscitation, which includes stimulation, and bag and mask ventilation immediately after birth is a key intervention for reducing neonatal morbidity and mortality [9]. Establishing effective ventilation within the Golden Minute —the first minute of life— is critical for neonates who fail to breathe spontaneously [10]. Failure to initiate spontaneous breathing can result can result in asphyxia, leading to hypoxic ischaemic encephalopathy, metabolic acidosis, neuronal injury, long term morbidity or death [11].

During bag and mask ventilation, monitoring the heart rate is essential to assess the effectiveness of the ventilation before deciding on the next course of action [10]. However, in most low-resource settings, human resources are limited and often only a single healthcare worker is available to perform the resuscitation [5]. This often necessitates pausing bag and mask temporarily to monitor the heart rate using a stethoscope or palpate for the heart rate at the umbilical cord stump. In addition, healthcare workers often clamp and cut the cord immediately, and move to a

designated neonatal resuscitation area which is away from the mother [12]. This practice hinders delayed cord-clamping and may also cause maternal anxiety and denial.

A potential is bedside neonatal resuscitation with an intact cord, facilitated by the BabySaver Tray and NeoBeat heart rate monitor. The BabySaver is a newborn resuscitation kit designed for bedside use [13], but not previously tested clinically, while the NeoBeat is a newborn heart rate meter that can be placed to the infant's torso to rapidly detect the heart rate and provide instant and continuous feedback to the healthcare worker [14]. Integrating these devices has the potential to enhance the process of bedside neonatal resuscitation with an intact cord, and reduce perinatal morbidity [15–20]. We hypothesized that using the BabySaver resuscitation platform and NeoBeat together could facilitate neonatal resuscitation with an intact cord in a low resource setting and improve outcomes. We therefore assessed the feasibility and acceptability of the combination in a regional hospital in Uganda.

## Methods

### Study design

This was a pre-post implementation study employing both quantitative and qualitative methods of data collection. The quantitative component involved non-participant observation of neonatal resuscitation practices, while the qualitative component explored health workers' and mothers' experiences with the BabySaver Tray and NeoBeat using a phenomenological approach.

### Study setting

We conducted this study between November 2023 and September 2024 at Mbale Regional Referral Hospital in Eastern Uganda. Mbale hospital is a government hospital serving a population of over 4–5 million people. The hospital conducts over 9,000 deliveries per year. Based on the annual medical records, about 6% of newborns require basic neonatal resuscitation. The hospital has one labour ward and one obstetric theatre with a single operating table. Babies are monitored during labour with intermittent auscultation using either a Pinard's stethoscope or handheld Doppler. Within the labour ward, resuscitation is done by the midwives who are trained in newborn resuscitation. The current practice in this facility is to transfer infants requiring resuscitation to a designated resuscitation area away from the mother and wrap the baby in blankets or towels as warm clothes to prevent hypothermia during resuscitation. Bedside resuscitation is occasionally done by individual midwives when a bag and mask is available near the bedside, and it is usually for milder cases where there is good tone but respiratory effort is considered inadequate. This is in response to scattered efforts to promote delayed cord clamping as recommended by WHO. After resuscitation, infants are transferred and admitted to the neonatal unit for assessment and ongoing management if needed. The hospital has a level II neonatal unit that is run by a neonatologist, six midwives and three clinical officers. The neonatal unit admits over 3600 infants annually, approximately half of these infants are inborn, and the leading cause of mortality is hypoxic ischaemic encephalopathy.

### Study population

We enrolled mother-infant dyads if they delivered in the labour ward or obstetric theatre at the Mbale Regional Referral Hospital and the infant was not breathing at birth, requiring neonatal resuscitation. We excluded all neonatal resuscitations conducted by a provider that had not been trained in the Helping Babies Breathe curriculum, as were infants born weighing <1.5 kg due to the NeoBeat size being unsuitable. Twins were excluded since the BabySaver Tray accommodates only one baby at a time. Infants with severe congenital anomalies, e.g., Downs syndrome, spina bifida, hydrocephalus, gastroschisis, and anencephaly) were also excluded. For the qualitative component, we interviewed healthcare workers who had used the BabySaver and the NeoBeat together for neonatal resuscitation and mothers whose babies had been resuscitated using these devices.

## Study procedures

The study was conducted in two phases. In the pre-implementation phase, we commenced with refresher training on neonatal resuscitation using the latest version of Helping Babies Breath (HBB) curriculum for all healthcare workers in the labour ward and obstetric theatre [21]. Certified neonatal resuscitation trainers from Karolinska Institutet provided the training. Once all healthcare workers had been trained, we observed their neonatal resuscitation practice for five months using a structured checklist explicitly developed for this study. In the implementation phase, we introduced the NeoBeat neonatal heart rate monitor and BabySaver. A team of master trainers from Haydom Lutheran Hospital, Tanzania, with expertise in the use of the NeoBeat trained the healthcare workers on use of the NeoBeat neonatal heart rate meter and BabySaver (Fig 1) for neonatal resuscitation. In the post-implementation phase, we then observed practice for six months using the same checklist. After two months of introducing the intervention, we commenced interviews with the healthcare workers and mothers to capture their experiences with the intervention.

## Sample size estimation and sampling

Based on the hospital's annual number deliveries (9,000), and the percentage of neonates (6%) that require resuscitation in Mbale Regional Referral Hospital (local data). We estimated that about 540 neonates would need resuscitation in a year (9000*0.06 = 540). Given the assumption that 50% of eligible neonates would receive bedside resuscitation (540*0.5 = 270). Over a period of six months, we aimed to observe (270/2 = 135). We inflated the sample size by 10 percent

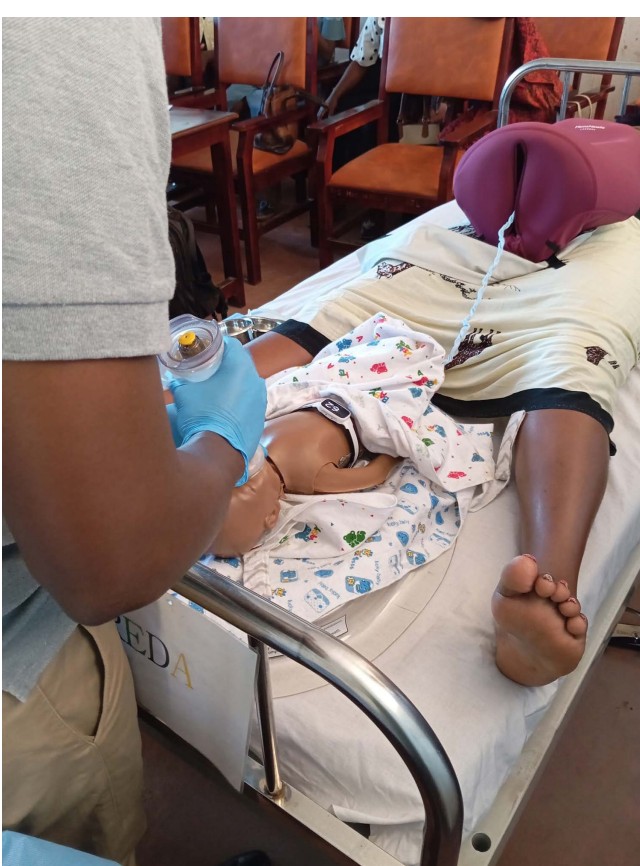

**Fig 1. BabySaver and NeoBeat setup for neonatal resuscitation.**

to cater for missed observations. The required sample size was 150 observations, with 75 in each phase. We assumed a two-sided confidence interval of 95%, and 99.9% power using a before and after approach. We consecutively enrolled the mother-infant dyads until the required sample size was accrued in each phase. As a feasibility study, no formal sample size calculation was needed, but we considered that a sample of 75 in each group would be adequate to detect an increase in bedside resuscitation from 10% baseline to 30% after implementation (5% significance level and 90% power).

For the qualitative interviews, participants were sampled purposively until data saturation was reached, leading to a total of 21 in-depth interviews with 10 health workers and 11 mothers. We purposefully selected participants with various experiences in neonatal resuscitation, and with variation in age and educational background.

## Variables

Our primary feasibility outcome was the proportion of infants requiring resuscitation who were resuscitated at the bedside using the BabySaver and NeoBeat together.

The explanatory variables included study phase, maternal sociodemographic factors such as age, marital status, education, residence, and occupation, as well as neonatal characteristics such as sex, birth weight, and gestational age.

Clinical outcome variables included APGAR scores at one and five minutes, neonatal temperature within the golden minute, commencement of resuscitation and 30 minutes after resuscitation at regular intervals of 10 minutes, outcome of the resuscitation process, and the condition of the infant at 24 and 48 hours after birth. We used an infra-red thermometer (Model:FT3010, Guangdong, China) to measure the infant's temperature.

## Data collection

For the quantitative component, we used the non-participant observation technique [22] to collect data on newborn resuscitation practice. Four trained research assistants who are qualified midwives collected data using a checklist designed in the KoBo toolbox (https://www.kobotoolbox.org/) on password protected smart tablet phones (Nokia T20, HMD Global Oy). The servers are secure and encrypted with strong safeguards and protection against data loss. We observed and noted the time of the different steps in the process of neonatal resuscitation using an application called 'Liveborn by LGH', developed by Laerdal Global Health. We used a checklist to collect data on mother and infant characteristics and clinical outcomes. To ensure completeness of the data, we reviewed maternal and infant medical records, maternity, and theatre registers to fill any gaps with the relevant information. The principal investigator reviewed the entries daily to ensure data quality and completeness.

In the qualitative data collection, healthcare workers who used the BabySaver and NeoBeat for neonatal resuscitation were invited to participate in an in-depth interview. Mothers whose infants were resuscitated using the BabySaver and the NeoBeat were also invited by the research assistant to participate in an in-depth interview shortly before their discharge from the hospital. The interviews were conducted by four research assistants; two male and two female study midwives trained in qualitative research data collection. We used semi-structured interview guides to collect data and these were used flexibly and modified as needed during the course of the study. We conducted interviews with the health workers two months into the post implementation phase. All the interviews with healthcare workers were conducted in English. We interviewed mothers after at least 24 hours after birth, at a time when they felt they were comfortable participating in the interview. Interviews with the mothers were conducted either in English or in their preferred local language (Luganda or Lumasaba or Ateso). All interviews were conducted face-to-face in a private room within the maternity unit The interviews lasted between 15–40 minutes and notes were taken during the interview. All the interviews were audiotaped, translated (as necessary) and transcribed by the study midwives.

## Data analysis

We analysed quantitative data in Stata version 18.0 (StataCorp; College Station, TX, USA). Data were compared descriptively, using frequencies and percentages for categorical variables and means, standard deviations, medians and

interquartile ranges for continuous variables. Pearson's Chi-square test or Fisher's exact test were used to test for proportion differences. The Mann Whitney U test was used to compare the group medians and the independent t-test was used to compare the group means. We fit generalized linear regression models of the binomial family with a log link and robust variance estimation to estimate prevalence ratios of various outcomes in the pre and post intervention period. The outcomes we studied included: time to cord clamping, outcome of resuscitation process, Apgar score, condition of the newborn at 24 hours post-resuscitation, condition of the newborn at 48 hours post-resuscitation. For each model we adjusted for the following confounders: antepartum hemorrhage and malpresentation. To estimate the change in temperature before, during and after resuscitation in the pre and post intervention periods, we fit a quadratic equation using Stata's *xtmixed* command and used the *margins* and *margin's* plot commands to graph the quadratic model. We used a quadratic equation because we did not expect a linear change in temperature over time.

Qualitative data were analysed using thematic analysis [23]. First, we read the transcripts and became familiar with the data. We then identified meaningful statements from phrases and sentences to generate the initial codes. Once the data had been sufficiently coded, the third step was to identify potential themes and sub-themes by combining all the relevant codes and data extracts into categories. We reviewed, modified themes and summarized our findings. Data analysis was done by MM and RN. We used Atlas ti.9 to organise the analysis process and borrowed upon Sekhon's acceptability model when presenting our findings. According to Sekhon, acceptability is a multi-faceted construct that reflects the extent to which people delivering or receiving a healthcare intervention consider it to be appropriate, based on anticipated or experienced cognitive and emotional responses to the intervention [24]. The theoretical framework of acceptability (TFA) consists of seven component constructs: affective attitude, burden, perceived effectiveness, ethicality, intervention coherence, opportunity costs, and self-efficacy [24].

### Ethical considerations

Ethical approval to conduct the study was obtained from the Busitema University Research and Ethics Committee, approval number; #REF BUFHS-164 and the Uganda National Council for Science and Technology (UNCST), #REF HS2676ES. Administrative clearance was granted by Mbale regional referral hospital. We obtained written informed consent from each participant before enrolling them into the study. For the quantitative component, informed consent was obtained in two steps as suggested in the differed consent pathway for intrapartum research [17]. Verbal consent was obtained from mothers during the early stages of labor to permit observation of neonatal resuscitation if required for their infant. Subsequently, written informed consent was obtained in the postpartum period to authorize the use of their data for the study.

## Results

### Study profile

Between November 2023 and September 2024, we screened a total of 280 neonatal resuscitations (149 in the pre-implementation phase and 131 in the post-implementation phase). During the pre-implementation period, 81 mothers-infant dyads were eligible, and 75 mothers consented to participate (Fig 2). In the post-implementation phase, 78 mothers-infant dyads were eligible, and 75 mothers consented to participate. The reasons for exclusion included neonatal resuscitations performed by a provider who had not been trained in the Helping Babies Breathe curriculum, infants born weighing <1.5 kg, twins, and infants with severe congenital anomalies.

### Sociodemographic characteristics of the participants

The participants recruited in the pre-and post-implementation phase had similar sociodemographic characteristics. Details are in Table 1.

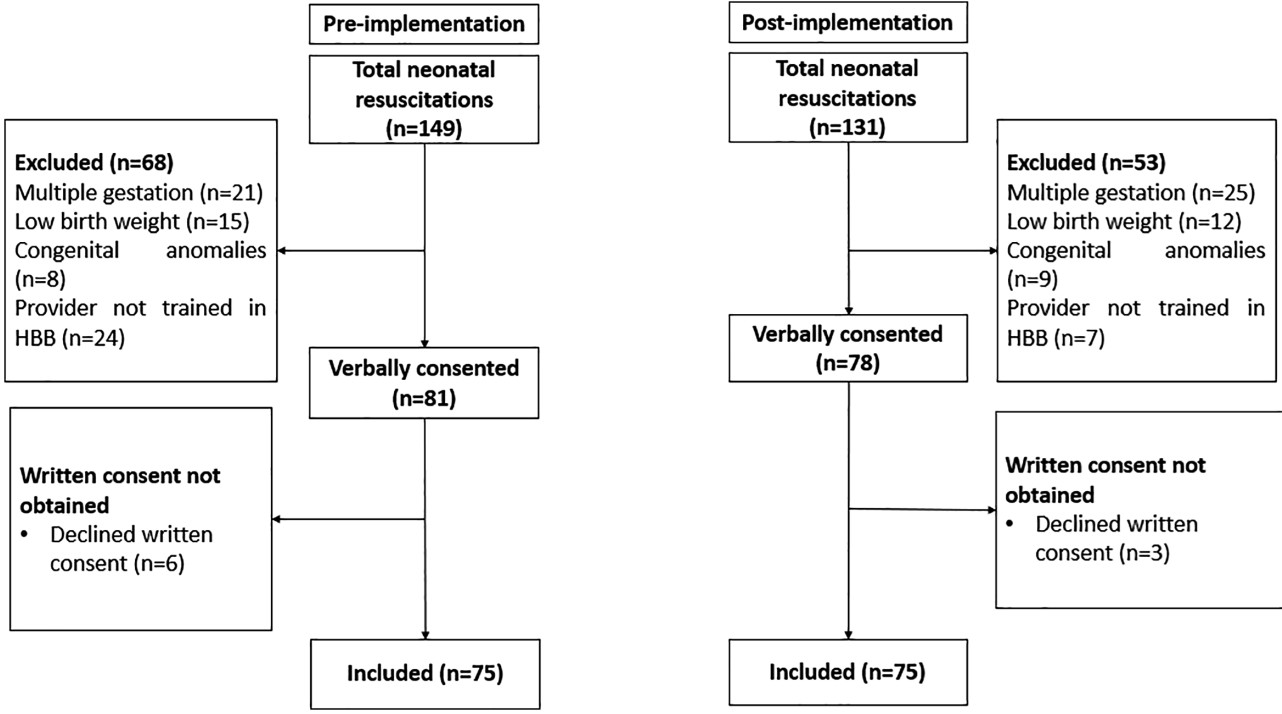

**Fig 2. Flow chart of the study participants.**

**Table 1. Sociodemographic characteristics of the study participants.**

| Variable | Pre-implementation N = 75 n (%) | Post-implementation N = 75 n (%) | Total N = 150 n (%) | P-value |
|---|---|---|---|---|
| Age (years) | | | | |
| <20 | 21 (28.0%) | 27 (36.0%) | 48 (32.0%) | 0.462 |
| 20–34 | 46 (61.3%) | 43 (57.3%) | 89 (59.3%) | |
| 35+ | 8 (10.7%) | 5 (6.7%) | 13 (8.7%) | |
| Marital status | | | | |
| Married | 8 (10.7%) | 14 (18.7%) | 22 (14.7%) | 0.166 |
| Single | 67 (89.3%) | 61 (81.3%) | 128 (85.3%) | |
| Education | | | | |
| None/Primary | 37 (49.3%) | 38 (50.7%) | 75 (50.0%) | 0.985 |
| Secondary | 30 (40.0%) | 29 (38.7%) | 59 (39.3%) | |
| Tertiary | 8 (10.7%) | 8 (10.7%) | 16 (10.7%) | |
| Occupation | | | | |
| Unemployed | 27 (36.0%) | 39 (52.0%) | 66 (44.0%) | 0.173 |
| Farmer | 24 (32.0%) | 14 (18.7%) | 38 (25.3%) | |
| Salaried employment | 11 (14.7%) | 11 (14.7%) | 22 (14.7%) | |
| Self-employment | 13 (17.3%) | 11 (14.7%) | 24 (16.0%) | |
| Residence | | | | |
| Rural | 51 (68.0%) | 49 (65.3%) | 100 (66.7%) | 0.729 |
| Urban | 24 (32.0%) | 26 (34.7%) | 50 (33.3%) | |

## Maternal and obstetric characteristics of the participants

The maternal and obstetric characteristics of the participants were also similar in both the pre- and post- implementation phases. Details are in Table 2.

## Proportion of infants resuscitated at the bedside at Mbale regional referral in Uganda

More infants were resuscitated at the bedside in the post-implementation period than in the pre-implementation period (34/75 versus 7/75, p<0.001). The mean time to initiating neonatal resuscitation was shorter in the post-implementation period as compared to the pre-implementation period (1.2 minutes versus 2.1 minutes, P=0.001). The median duration of neonatal resuscitation in the post-implementation period was shorter than in the pre-implementation period (5 minutes versus 8 minutes, p<0.001). Details are in Table 3.

## Newborn temperature variation before and after resuscitation in the pre- and post-implementation period

Newborns in the post-implementation period had higher axillary temperatures at birth, and at 0-, 10-, 20-, and 30-minutes post resuscitation as compared to those in the pre-implementation period. Details are in Fig 3.

## Comparison of timing of cord clamping and birth outcomes pre- and post- implementation of bedside neonatal resuscitation at Mbale regional referral hospital in Uganda

**Timing of cord clamping.** Health workers were 13.58 times more likely to clamp the cord after 3 minutes in the post-implementation than in the pre-implementation phase (aPR: 13.58; 95% CI: 3.35–55.09). Details are in Table 4.

**Outcome of the resuscitation process.** Newborns in the post-implementation period were 9.27 times more likely to be alive and well (transferred to the postnatal ward with the mother) (aPR: 9.27; 95% CI: 2.23–38.48) and 34% less likely to be alive and not well (transferred to the neonatal unit) (aPR: 0.66; 95% CI: 0.56–0.78) after the resuscitation process than those in the pre-implementation period.

**Apgar score at 5 minutes.** Newborns in the post-implementation period were 64% less likely to have an APGAR score less than 7 at 5 minutes than those in the pre-implementation period (aPR: 0.36; 95% CI: 0.26–0.50).

There was no difference in the condition of the infant at 24 and 48 hours between the pre- and post- implementation periods.

## Acceptability of using the BabySaver and NeoBeat together for bedside neonatal resuscitation

**Participant characteristics.** A total of 21 participants were recruited; 10 health workers and 11 mothers. The median age of the health workers was 33.5 (31-44) and their average working experience was 13.4 years. Mothers' age ranged from 17 to 39 years. Most of the mothers, (72.7%, n=8) were married and (46.7%, n=7) had secondary education level.

We presented health workers' and mothers' views and experiences with the use of BabySaver and NeoBeat together for bedside neonatal resuscitation using Sekhon's theoretical framework of acceptability [24] (S1 Table).

## A. Affective attitude

## BabySaver and NeoBeat use simplifies neonatal resuscitation

Health workers reported that use of the BabySaver and NeoBeat simplified the practice of neonatal resuscitation as they are able to perform the resuscitation successfully, even without an assistant.

> *"This new era of Neobeat and resuscitation platform, it can be done without an assistant. It's even easier and not cumbersome"* HW 008

**Table 2. Maternal and obstetric characteristics of the study participants.**

| Variable | Pre- implementation N = 75 n (%) | Post-implementation N = 75 n (%) | Total N = 150 n (%) | P-value |
|---|---|---|---|---|
| Gestation | | | | |
| Preterm (≤36 + 6 weeks) | 14 (18.7%) | 17 (22.7%) | 31 (20.7%) | 0.545 |
| Term | 61 (81.3%) | 58 (77.3%) | 119 (79.3%) | |
| Gravidity | | | | |
| Primigravida | 33 (44.0%) | 36 (48.0%) | 69 (46.0%) | 0.848 |
| Parous | 30 (40.0%) | 29 (38.7%) | 59 (39.3%) | |
| Grand multigravida (≥5 previous births at > 28 weeks) | 12 (16.0%) | 10 (13.3%) | 22 (14.7%) | |
| History of pregnancy loss (any gestation) | | | | |
| Yes | 12 (28.6%) | 10 (25.6%) | 22 (27.2%) | 0.767 |
| No | 30 (71.4%) | 29 (74.4%) | 59 (72.8%) | |
| Type of pregnancy loss | | | | |
| Abortion/Miscarriage | 11 (91.7%) | 9 (90.0%) | 20 (90.9%) | 0.892 |
| Stillbirth | 1 (8.3%) | 1 (10.0%) | 2 (9.1%) | |
| Mother admitted as a referral in labour | | | | |
| Yes | 43 (57.3%) | 48 (64.0%) | 91 (60.7%) | 0.403 |
| No | 32 (42.7%) | 27 (36.0%) | 59 (39.3%) | |
| Source of the referral | | | | |
| Government facility | 43 (100.0%) | 46 (95.8%) | 89 (97.8%) | 0.176 |
| Private facility | 0 (0.0%) | 2 (4.2%) | 2 (2.2%) | |
| Herb use during the labour | | | | |
| Yes | 6 (8.0%) | 6 (8.0%) | 12 (8.0%) | 1.000 |
| No | 69 (92.0%) | 69 (92.0%) | 138 (92.0%) | |
| Obstetric complications | | | | |
| Antepartum hemorrhage | 6 (8.0%) | 1 (1.3%) | 7 (4.7%) | 0.053 |
| Preterm labour | 5 (6.7%) | 4 (5.3%) | 9 (6.0%) | 0.731 |
| Hypertensive disorder | 9 (12.0%) | 12 (16.0%) | 21 (14.0%) | 0.480 |
| Obstructed labour | 24 (32.0%) | 25 (33.3%) | 49 (32.7%) | 0.862 |
| Prolonged labour | 7 (9.3%) | 5 (6.7%) | 12 (8.0%) | 0.547 |
| Malpresentation | 2 (2.7%) | 8 (10.7%) | 10 (6.7%) | 0.050 |
| Other | 40 (53.3%) | 29 (38.7%) | 69 (46.0%) | 0.072 |
| Medical conditions | | | | |
| Diabetes Mellitus | 0 (0.0%) | 1 (1.3%) | 1 (0.7%) | 0.316 |
| Hypertension | 0 (0.0%) | 1 (1.3%) | 1 (0.7%) | 0.316 |
| HIV | 2 (2.7%) | 2 (2.7%) | 4 (2.7%) | 1.000 |
| Malaria | 1 (1.3%) | 0 (0.0%) | 1 (0.7%) | 0.316 |
| Other | 8 (10.7%) | 4 (5.3%) | 12 (8.0%) | 0.229 |
| APGAR score at 1 min | | | | |
| <7 | 74 (98.7) | 69 (92.0) | 143 (95.3) | 0.053 |
| ≥7 | 1 (1.3) | 6 (8.0) | 7 (4.7) | |
| Sex of the baby | | | | |
| Male | 39 (52.0%) | 43 (57.3%) | 82 (54.7%) | 0.512 |
| Female | 36 (48.0%) | 32 (42.7%) | 68 (45.3%) | |
| Weight of the baby | 3.1 (2.7-3.3) | 3 (2.79-3.4) | 3.1 (2.8-3.4) | |
| <2.5 kg | 15 (20.0%) | 11 (14.7%) | 26 (17.3%) | 0.599 |
| 2.5 to 4.0 | 57 (76.0%) | 62 (82.7%) | 119 (79.3%) | |
| >4.0 kg | 3 (4.0%) | 2 (2.7%) | 5 (3.3%) | |

**Table 3. Newborns resuscitated at the bedside in the pre and post implementation phase of the study.**

| Study phase | Place of resuscitation | | | | p-value |
|---|---|---|---|---|---|
| | Bedside N=41 n (%) | Designated area N=100 n (%) | Both N=9 n (%) | Total | |
| Pre-implementation | 7 (17.1%) | 68 (68.0%) | 0 (0.0%) | 75 (50.0%) | <0.001 |
| Post-implementation | 34 (82.9%) | 32 (32.0%) | 9 (100%) | 75 (50.0%) | |

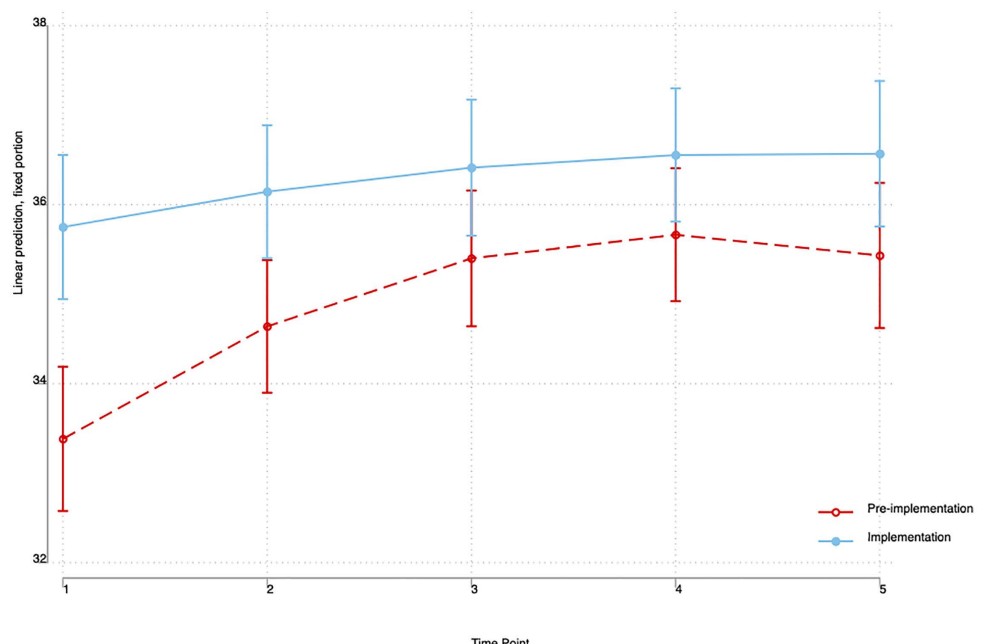

**Fig 3. Trends in the axillary temperatures of newborns at birth, during, and after resuscitation.** Legend: Time 1 = temperature within the golden minute, Time 2 = temperature at zero minutes after resuscitation, Time 3 = temperature at 10 minutes after resuscitation, Time 4 = temperature at 20 minutes after resuscitation, Time 5 = temperature at 30 minutes after resuscitation.

*"… so to my experience, I see the Neobeat and the BabySaver are making my work like simpler than before…..*
*Because I don't have to start again touching the stethoscope to listen to the heartbeat. So, I'll be focusing on my resuscitation as I'm looking at the readings from the NeoBeat." HW 004*

## Resuscitation at the bedside with an intact cord is time saving

Health workers appreciated the fact that bedside resuscitation ensures timely provision of neonatal resuscitation as they do not waste time cutting the cord to transfer the baby to a designated resuscitation area.

*"I felt it was so easy. Because, first of all, it saves time. And I don't waste time to tie the cord and transfer the baby to the other side." HW 007*

*"But now, as they brought this... the BabySaver...It is easy. It is just done within there. you don't lose any minute." HW 005*

**Table 4. Comparison of timing of cord clamping and birth outcomes pre- and post- implementation of bedside neonatal resuscitation in Uganda.**

| Variable | Pre-implementation N=75 n (%) | Post-implementation N=75 n (%) | Total N=150 n (%) | cPR* | 95% CI | P-value | aPR# | 95% CI | P-value |
|---|---|---|---|---|---|---|---|---|---|
| Time to cord clamping (in minutes) | | | | | | | | | |
| Median (IQR) | 1 (0–2) | 2 (1–6) | 1 (1–3) | | | | | | |
| <1 | 20 (26.7%) | 9 (12.0%) | 29 (19.3%) | 0.45 | 0.22–0.93 | 0.030 | 0.48 | 0.23–0.97 | 0.042 |
| 1-3 | 53 (70.7%) | 37 (49.3%) | 90 (60.0%) | 0.70 | 0.53–0.92 | 0.010 | 0.52 | 0.40–0.67 | <0.001 |
| >3 | 2 (2.7%) | 29 (38.7%) | 31 (20.7%) | 14.5 | 3.57–58.89 | <0.001 | 13.58 | 3.35–55.09 | <0.001 |
| Outcome of the resuscitation process | | | | | | | | | |
| Alive and well | 2 (2.7%) | 19 (25.3%) | 21 (14.0%) | 9.5 | 2.28–39.55 | 0.002 | 9.27 | 2.23–38.48 | 0.002 |
| Alive and not well | 71 (94.7%) | 53 (70.7%) | 124 (82.7%) | 0.69 | 0.59–0.79 | <0.001 | 0.66 | 0.56–0.78 | <0.001 |
| Fresh Still birth (no heart rate present despite resuscitation) | 2 (2.7%) | 0 (0.0%) | 2 (1.3%) | – | – | – | – | – | – |
| Early neonatal death (heart rate detected, but no spontaneous breathing) | 0 (0.0%) | 3 (4.0%) | 3 (2.0%) | – | – | – | – | – | – |
| Apgar score <7 at 5 minutes | 45 (60.0%) | 25 (33.3%) | 70 (46.7%) | 0.56 | 0.38–0.80 | 0.002 | 0.36 | 0.26–0.50 | <0.001 |
| Condition of the Newborn at 24 hours post-resuscitation*** | | | | | | | | | |
| Alive and well no HIE | 50 (68.5%) | 57 (79.2%) | 107 (73.8%) | 1.14 | 0.93–1.39 | 0.211 | 1.14 | 0.93–1.40 | 0.212 |
| HIE¥ | 14 (19.2%) | 12 (16.7%) | 26 (17.9%) | 0.86 | 0.42–1.73 | 0.668 | 0.82 | 0.39–1.71 | 0.590 |
| Dead | 9 (12.3%) | 3 (4.2%) | 12 (8.3%) | 0.33 | 0.09–1.19 | 0.090 | 0.37 | 0.09–1.53 | 0.170 |
| Condition of the Newborn at 48 hours post-resuscitation*** | | | | | | | | | |
| Alive and well no HIE | 50 (68.5%) | 56 (77.8%) | 106 (73.1%) | 1.12 | 0.91–1.38 | 0.286 | 1.12 | 0.91–1.38 | 0.293 |
| HIE | 11 (15.1%) | 12 (16.7%) | 23 (15.9%) | 1.09 | 0.51–2.32 | 0.821 | 1.04 | 0.47–2.33 | 0.917 |
| Dead | 12 (16.4%) | 4 (5.6%) | 16 (11.0%) | 0.33 | 0.11–0.99 | 0.048 | 0.36 | 0.11–1.16 | 0.087 |

* cPR – crude prevalence ratio.

# aPR – adjusted prevalence ratio.

¥ HIE – Hypoxic Ishaemic Encephalopathy.

*** Did not include the Fresh Still Births and Early Neonatal Deaths at birth.

Further, they also noted that the BabySaver provides a clean environment for resuscitation.

*"You know the BabySaver provides a clean environment for resuscitation. Because you cover with a sheet, you put the baby, and then you cover on top and just expose the chest. You have covered the baby's head." HW 001*

## Bedside resuscitation builds trust between health workers, patients and caregivers

Participants also highlighted that resuscitating at the bedside enhances trust between the health workers and the patients as they are able to see the process of the resuscitation.

*"For me as a health worker, I advocate for it because many of us have gone…. to court because of false accusations. They exchanged my baby. They have killed my baby. Maybe Musawo (health worker) was not minding. So, when they are there, I think that it's also better the health worker. For me, if it is me, whether you consent or not, I have to resuscitate from there because I need to safeguard myself." HW 010*

*"I don't think it's a challenge because the mother gets to know what is going on. Like in, even if the baby dies, like she will not say she was neglected, Musawo (health worker) tried." HW 004*

### NeoBeat use during resuscitation motivates the health worker

Health workers reported that using the NeoBeat guided the process of resuscitation and motivated them to perform better.

*You can't lose hope. You see the heart rate showing, improving. So, you say, the baby is improving. I should not give up. So, it's really very good. HW 001*

*It will help you to know whether your resuscitation is a success or you should abandon it. If you are resuscitating and you see the fetal heart is improving, decreasing, it motivates you to say, I think this baby is still alive. HW 006*

### Reassurance to the mother

Participants also noted that resuscitating at the bedside reassures the mother about their baby's condition.

*"I feel good because while the mother is observing the baby, she is reassured that the baby is going to be fine than taking away the baby to some other place." HW 007*

*"Yes. Because it gives you a sense of... Saying my baby is coming slowly, slowly. Even if things don't turn out good, you'd be okay that that was my baby. Yeah. They have not changed, they have not done anything wrong. It was God's plan. Okay." Mother 002*

### Promotes patient satisfaction

Another perceived benefit was that bedside resuscitation promotes patient satisfaction irrespective of the outcomes.

*"There is that kind of the relationship between the health worker and the mother because… (why do I say so?) … the mother really will see you struggling and the baby came up. Of course, the mother will have to say thank you." HW 003*

### Coping with loss is smoother

They also perceived that coping with loss was smoother as compared to when the baby is resuscitated away from the mother.

*"Even if the baby dies, actually breaking out the news, it becomes so easier for you compared when you resuscitate this baby in a very far place or in a different way. And even in a bereavement of a mother, she will start building up from when she's observing how the baby is trying to cope up with life. And I think in bereavement it becomes easier for the mother to cope up and say, they tried to do this ABCD and the baby failed." HW 010*

### Bedside resuscitation motivates the health worker

Health workers noted that they are motivated to do their best because the mother is observing the process of the resuscitation.

*"You always feel guilty to do your best because the mother is observing." HW 003*

## B. Burden and self-efficacy

### The BabySaver and NeoBeat are easy to use

Health workers commented that using of the BabySaver for neonatal resuscitation is easy and does not require any technical skills.

*"This new era of NeoBeat and resuscitation platform, it can be done without an assistant. It's even easier and not cumbersome" HW 008*

*"It doesn't need a lot of skills, ….. It is flat, and it has that position for the head where we can put the baby in a sniffing position to make the airway pattern such that you are able to bag and you are able, the baby is able to take in what? The breath that you are delivering." HW 009*

In addition, they reported that applying the NeoBeat to the newborn's torso is very easy and appreciated the fact that it is portable and wireless.

*"Putting the NeoBeat is very easy because we able to get the baby's heart rate very fast without looking for the pericordium or the cord for us to start looking out to count the heart rate." HW 008*

*"It is a portable device, which is very good. And it is easy to attach and detach if needed. And also, it has, it is wireless, you know? Like that is easier, you don't need to connect it to power when you are resuscitating." HW 008*

## C. Intervention coherence and perceived effectiveness

### BabySaver provides a flat surface for resuscitation

Health workers found the BabySaver beneficial as it provides a stable, and flat surface for effective neonatal resuscitation

*"it gives us that flat surface whereby you can place (the baby) and it has a design whereby even you can put the head of the baby for positioning. It mainly provides that surface for easy maneuvering." HW 003*

### Bedside resuscitation is effective: improved health outcomes

Bedside resuscitation was perceived as effective in reducing the need for neonatal unit admissions.

*"It is really effective. The babies cry after some time, after resuscitation. And even, most of them we don't send them to the neonatal unit. They pick up very early." HW 001*

## D. Opportunity costs and Ethicality

### Cultural beliefs, myths and misconceptions: Mothers blood attacks the baby, Fear of HIV transmission:

Some participants believed that the mother's blood can attack the baby and the baby dies if the cord is not cut immediately.

*"Yeah. You're supposed to cut immediately. Even though that the last one (placenta) has not come, they need to cut the cord faster. Quickly, quickly. That they don't want blood of the mother to attack the baby." Mother 002*

*"Some people say that the baby dies in case you don't cut the cord immediately. There in the villages even in the health facilities they say that once the baby is born and cries you have to cut the cord immediately otherwise the baby might die."* Mother 012

*"But it will depend on which patient you are dealing with. Because for a mother with HIV, I cannot do it, I cannot wait for all that time. This baby should be separated immediately from the mother, and to avoid the vertical transmission of HIV/AIDS."* HW 009

### Fear of hypothermia: Covering the baby isn't enough

Despite efforts to keep newborns warm by covering with a cloth during the resuscitation, health workers noted that covering alone is insufficient, as babies often require additional warming after birth.

*"The worst part of it is I think the warmth, but the rest is okay. Even if we cover. They come up from hibernation so they need a warmer. We try to cover but still...That's the only thing maybe."* HW 002

### Perceived risk for cross infections

Some health workers expressed concerns about the effectiveness of alcohol in eliminating microbes.

*"Yes, alcohol is very good because it dries off very fast. But however, I don't know whether it kills all the microbes on it. Like I said, some other babies come up with different Meconium. Meconium, yes, chorioamnionitis, so that's where my mind can hesitate"* HW 003.

### Difficult to use the BabySaver in theatre

Health workers also reported that the limited space in the operating theatre and positioning constraints make use of the BabySaver challenging in theatre.

*"Barriers is there is no space. The space is limited. Now, it might even contaminate the area of the surgery."* HW 003

*"As I told you, in theatre, we still have a challenge. Implementing it the way we use it in the labor ward. Because the mother is in a recumbent position. They tie somehow the legs. They want them to be together."* HW 001

Another health worker highlighted the challenge of visualization of the resuscitation by the mother, as sterility requirements prevent mothers from directly seeing what is happening to their baby (Table 5).

*"And here, even if you have performed a spinal anaesthesia, they are not able to see. But you only explain to them, you know, the baby has come out but is not doing well and we are trying our best to make sure the baby improves on this BabySaver. I would like the mother to see what is going on. But due to the nature of the procedure we need the environment to be sterile."* HW 009

## Discussion

This study was aimed at assessing the feasibility and acceptability of using the BabySaver and NeoBeat together for bedside neonatal resuscitation. Our findings show that it is feasible to use the BabySaver and NeoBeat together for bedside

**Table 5. Suggested modifications for the BabySaver.**

| Attribute | Quote |
|---|---|
| Size | "Yes, looking at the size, like I commented earlier, and I was looking at the size. The size, also, I'm looking at it is a bit bigger. If we can make something that can fit the baby." HW 003<br>"If there is a provision that they can provide for small babies and premature, then the bigger babies, it's fine. But you never know." HW 009 |
| Shape | "No, like making it flatter. How do you call that? Making it a bit harder, like flatter with no much elevations." HW 008 |
| Edges | "Yes, and then I'm looking at trauma to the mother. When I checked it, I examined it down, I'm imagining if it pressed very hard, you know when you're resuscitating the baby, it can easily injure the mother, the lower limbs. And from my experience in the theatre, these mothers were given spinal (anaesthesia), they may not know feel, but the injury will be felt later. So that it has to be blunt a bit, blunt on down surface. So that however much I press it, it cannot injure the mother, mother's laps." HW 003 |
| Material | "And when I look at this material, it's not that metallic. But if we had a material that can be autoclavable, we'd autoclave it. And maybe it's part of the Caesar sets. Whenever we're preparing for Caesar set, whenever I'm preparing for SVD, it is there. It is, that one can be easy for me. Then this business of I want to wrap it, I'm struggling to wrap it with a sterile cloth. But if it can be like any other resuscitation platform material that I can sterilize, that would be very good" HW 003 |
| Heating mechanism | "Maybe if they accompany it with some kind of heat. If there can be a provision for heat, it can be the best. Most of the time we fear for hypothermia when you are resuscitating in the mother's legs." HW 008 |
| Cleaning and disinfection | "I think it will help because if you sterilize it, it is sterile. It creates a sterile place for the baby to prevent some infections." HW 006 |
| Use in theatre | "But here in the theatre, the best is on top of the limbs. You can just separate them a bit and put on the baby saver." HW 009 |

neonatal resuscitation. More babies were resuscitated at the bedside with an intact cord, the time to achieve successful neonatal resuscitation was much shorter and fewer babies needed further care post resuscitation. In addition, babies resuscitated at the bedside with an intact cord had a higher axillary body temperature. The use of these devices for bedside neonatal resuscitation was also acceptable to both the health workers and the mothers. Being able to resuscitate at the bedside was reassuring to the mother, improved trust and promoted patient satisfaction irrespective of the outcomes. However, there were myths and misconceptions about delayed cord clamping, concerns about hypothermia, perceived risk for cross infections and difficult use in theatre that may hinder BabySaver and NeoBeat use in this setting.

### Feasibility of using the BabySaver and the NeoBeat together for neonatal resuscitation

The increase in the number of babies resuscitated at the bedside observed in our study was not surprising as previous studies [25–27] have highlighted similar findings. Availing a mobile, firm and flat surface enhances the practice of initiating neonatal resuscitation at the bedside with an intact cord (delayed cord clamping), within the first golden minute [13]. This is exactly what the BabySaver provides in resource limited settings such as this one. Furthermore, we observed that the time taken to achieve successful resuscitation was much shorter after the intervention was introduced (BabySaver and Neobeat). This could be explained by the fact that health workers were able to initiate neonatal resuscitation within the golden minute, as they didn't encounter time delays related to cutting the cord and moving to a designated resuscitation area. In addition, babies resuscitated with an intact cord continue receiving oxygenated blood from the placenta, which results in an increased pulmonary blood flow, and earlier initiation of breathing [18–20,28]. In addition, the NeoBeat provides continuous newborn heart rate monitoring during the resuscitation, guides the process of resuscitation and enables the health worker to concentrate on Bag and Mask Ventilation [29]. Therefore, it's not surprising that we noted fewer newborn morbidities in the post-implementation period. Several studies

including systematic reviews have demonstrated the benefits of newborn resuscitation with an intact cord provides several new-born benefits such as life support, which is associated with improved newborn outcomes [18,19,27]. Further, the slightly different training for staff in the pre and post intervention period may have influenced the outcomes. However, we did not observe differences in the condition of the infant at 24 and 48 hours. This may be due to the limitations of short-term outcome measures, which may not adequately reflect the benefits of timely bedside resuscitation.

Contrary to the commonly held belief that resuscitation at the bedside without a warmer exposes these newborns to the risk of hypothermia, we observed that infants resuscitated at the bedside with an intact cord had a higher axillary body temperature both before and after the resuscitation (Fig 1). This could be related to the fact these infants achieve improved perfusion of organs and peripheral tissue including skin and muscles due to optimized blood volume and pressure from the placenta at the same body temperature with the mother [30]. Additionally, successful resuscitation being achieved in a short time could have contributed to the quick stabilization of the body temperatures of the infants. As such, it may not be necessary to incorporate any heating mechanism in the future design of the BabySaver, which would help keep the production costs of the BabySaver to a minimum.

## Acceptability of using the BabySaver resuscitation platform and the NeoBeat together for neonatal resuscitation

Keeping the baby and the mother together during resuscitation was reassuring to the mother, improved trust and promoted patient satisfaction irrespective of the outcomes. This is consistent with previous studies [20,27] that have reported benefits of bedside neonatal resuscitation. This is important most especially in low resource settings where there are cases of litigations arising from allegations of negligence or switching babies in case of poor outcomes or when the baby dies [31,32].

The BabySaver and the NeoBeat simplified the practice of neonatal resuscitation because they were easy to use, mobile and lightweight. These attributes have been reported in previous studies where these devices have been used [29]. This could potentially inform the scale up of these devices for newborn resuscitation with an intact cord in low resource settings where this practice is hindered by limited availability of the right equipment.

Our study also revealed myths and misconceptions about delayed cord clamping such as mother's blood attacking the baby and HIV transmission were voiced by the participants. This may be related to sociocultural meanings attached to childbirth and the umbilical cord in this setting [33,34]. Similarly, a study in Tanzania that explored experiences of skilled birth attendants with early or delayed cord clamping revealed that some health workers were concerned about practicing delayed cord clamping among HIV positive mothers [35]. However, existing evidence shows that HIV cannot be transmitted through delayed cord clamping [35,36]. The BabySaver and the NeoBeat are designed to be cleaned using 0.5% Chlorine water and or 99.9 percent alcohol. However, some health workers falsely perceived this was inadequate and would increase the risk for cross infections. This underscores the need for health education and sensitization of mothers and health workers to dispel myths and misconceptions about delayed cord clamping and infection prevention.

Use of the intervention in theatre was challenging because of fear of contaminating the surgical field, positioning the BabySaver, and inability of the mother to visualize the resuscitation. This was raised despite the fact that we provided autoclavable linens to cover the BabySaver when being used in theatre on top of the mother's legs. Addressing these issues may require modifications to the BabySaver's design, such as making it attachable to existing theatre equipment without interfering with sterile areas. Further research is needed to explore optimal positioning strategies and usability of the BabySaver in theatre.

Respondents suggested a number of modifications to the design of the BabySaver such as the need to reduce the size of the BabySaver to fit on the delivery beds. Most of the deliveries are conducted in a dorsal lithotomy position on a standard delivery bed which is much smaller than in other settings where mothers can deliver in upright positions while squatting or use a birthing stool and the BabySaver can be placed on the floor [37,38]. In addition, there were suggestions to make the ridge on the BabySaver less prominent as it hyperextends the newborn's head during resuscitation. This is

contrary to the current teaching of HBB which emphasizes the need to prevent hyperextension of the neck during mask ventilation as it causes airway obstruction [39,40]. These changes will be factored into the ongoing modifications to the BabySaver designs to improve its usability for neonatal resuscitation.

### Strengths and limitations

One of the strengths of this study is that it's the first study to assess the combined use of the BabySaver and NeoBeat for neonatal resuscitation. In addition, we utilized a mixed methods approach which enhances the rigor, and trustworthiness of our findings. Furthermore, in the qualitative component, we captured both the views of the health workers and the mothers regarding the use of the BabySaver and NeoBeat for neonatal resuscitation. Nonetheless, our quantitative component was limited by the use of a quasi-experimental study design which is prone to bias [41]. However, we believe the time interval between the pre- and post-implementation phases were too short for factors other than the intervention to have caused the observed effects. Additionally, the researchers who conducted the interviews were involved in the implementation of the BabySaver and NeoBeat for neonatal resuscitation in the labour ward, which may have biased the views from the participants. However, we mitigated bias by training researchers on qualitative data collection techniques including the practice of reflexivity. One limitation of infrared temperature assessment is the tendency to underestimate readings in the presence of moisture. We took our first temperature measurement immediately after birth, before drying the baby. This could have resulted in a systematic underestimation of the temperature reading at 1 minute. However, this error would have been similar in both the pre-implementation and post-implementation period and we suspect that this did not affect the difference between the two periods.

### Conclusion

Use of the BabySaver and NeoBeat for neonatal resuscitation improved the practice of bedside resuscitation and reduced neonatal morbidity. It was also acceptable to health workers and mothers as it provided reassurance, improved trust and promoted patient satisfaction irrespective of the outcomes. The scale up should include strategies to dispel myths and misconceptions about delayed cord clamping and developing optimal approaches to promote its usability in theatre.

### Supporting information

**S1 Table. A summary of results on acceptability of using the BabySaver and NeoBeat together for bedside neonatal resuscitation, using the Sekhon's model of acceptability.** TFA = Theoretical Framework on Acceptability. (DOCX)

### Author contributions

**Conceptualization:** Milton W. Musaba, Ritah Nantale, David Mukunya, Julius N. Wandabwa, Kathy Burgoine, Kenneth Mugabe, Thorkild Tylleskar, Andrew D. Weeks.

**Data curation:** Milton W. Musaba, David Mukunya, Julius N. Wandabwa, Kathy Burgoine, Nicolas J. Pejovic, Benjamin A. Kamala, Agnes Napyo, Brendah Nambozo, Faith Oguttu, John Stephen Obbo, Thorkild Tylleskar, Andrew D. Weeks.

**Formal analysis:** Milton W. Musaba, Ritah Nantale, David Mukunya.

**Funding acquisition:** Milton W. Musaba.

**Investigation:** Milton W. Musaba, Ritah Nantale, David Mukunya, Julius N. Wandabwa, Kathy Burgoine, Agnes Napyo, Kenneth Mugabe, Brendah Nambozo, Faith Oguttu, Thorkild Tylleskar.

**Methodology:** Milton W. Musaba, Ritah Nantale, David Mukunya, Julius N. Wandabwa, Kathy Burgoine, Nicolas J. Pejovic, Benjamin A. Kamala, Agnes Napyo, Kenneth Mugabe, Brendah Nambozo, Faith Oguttu, John Stephen Obbo, Thorkild Tylleskar, Andrew D. Weeks.

**Project administration:** Milton W. Musaba, Julius N. Wandabwa, Kathy Burgoine, Kenneth Mugabe, John Stephen Obbo, Thorkild Tylleskar.

**Resources:** Nicolas J. Pejovic, Benjamin A. Kamala, Kenneth Mugabe, John Stephen Obbo.

**Software:** Ritah Nantale.

**Supervision:** Milton W. Musaba, David Mukunya, Kathy Burgoine.

**Validation:** Milton W. Musaba, Ritah Nantale, David Mukunya, Julius N. Wandabwa, Kathy Burgoine, Nicolas J. Pejovic, Benjamin A. Kamala, Agnes Napyo, Kenneth Mugabe, Brendah Nambozo, Faith Oguttu, John Stephen Obbo, Thorkild Tylleskar, Andrew D. Weeks.

**Visualization:** Ritah Nantale, David Mukunya.

**Writing – original draft:** Milton W. Musaba, Ritah Nantale, David Mukunya, Andrew D. Weeks.

**Writing – review & editing:** Milton W. Musaba, Ritah Nantale, David Mukunya, Julius N. Wandabwa, Kathy Burgoine, Nicolas J. Pejovic, Benjamin A. Kamala, Agnes Napyo, Kenneth Mugabe, Brendah Nambozo, Faith Oguttu, John Stephen Obbo, Thorkild Tylleskar, Andrew D. Weeks.

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
