## [Decision Letter · Decision Letter 0]

23 Jun 2025

Dear Dr. Nantale,

Thank you for submitting your manuscript to PLOS ONE. After careful consideration, we feel that it has merit but does not fully meet PLOS ONE’s publication criteria as it currently stands. Therefore, we invite you to submit a revised version of the manuscript that addresses the points raised during the review process.

**Please address all comments from both reviewers, separately, as described below.**
**In addition, please address the following:**
https://safri.org/the-baby-saver/

Please submit your revised manuscript by Aug 07 2025 11:59PM.If you will need more time than this to complete your revisions, please reply to this message or contact the journal office at plosone@plos.org . A rebuttal letter that responds to each point raised by the academic editor and reviewer(s). You should upload this letter as a separate file labeled 'Response to Reviewers'.A marked-up copy of your manuscript that highlights changes made to the original version. You should upload this as a separate file labeled 'Revised Manuscript with Track Changes'.An unmarked version of your revised paper without tracked changes. You should upload this as a separate file labeled 'Manuscript'.

We look forward to receiving your revised manuscript.

Kind regards,

Alan Richard Horn, MD, DCH, FCPaed, Cert. Neonatology, PhD

Academic Editor

PLOS ONE

Journal Requirements:

2. Please include a complete copy of PLOS’ questionnaire on inclusivity in global research in your revised manuscript. Our policy for research in this area aims to improve transparency in the reporting of research performed outside of researchers’ own country or community. The policy applies to researchers who have travelled to a different country to conduct research, research with Indigenous populations or their lands, and research on cultural artefacts. The questionnaire can also be requested at the journal’s discretion for any other submissions, even if these conditions are not met.  Please find more information on the policy and a link to download a blank copy of the questionnaire here: https://journals.plos.org/plosone/s/best-practices-in-research-reporting. Please upload a completed version of your questionnaire as Supporting Information when you resubmit your manuscript.”

[This study was supported through the Saving Lives at Birth program in Low and Middle Income Countries under the Laerdal Foundation (#REF:2022-0264) The funding sources had no role in the study design, data collection, data analysis, data interpretation, the manuscript's writing, or the decision to submit the manuscript for publication. The content is solely the responsibility of the authors and does not necessarily represent the official views of the funding sources.].

4. We notice that your supplementary tables are included in the manuscript file. Please remove them and upload them with the file type 'Supporting Information'. Please ensure that each Supporting Information file has a legend listed in the manuscript after the references list.

Reviewers' comments:

Reviewer's Responses to Questions

**Comments to the Author**

1. Is the manuscript technically sound, and do the data support the conclusions?

Reviewer #1: Yes

Reviewer #2: Yes

2. Has the statistical analysis been performed appropriately and rigorously?

Reviewer #1: Yes

Reviewer #2: Yes

3. Have the authors made all data underlying the findings in their manuscript fully available?

Reviewer #1: Yes

Reviewer #2: Yes

4. Is the manuscript presented in an intelligible fashion and written in standard English?

Reviewer #1: Yes

Reviewer #2: Yes

Reviewer #1: Title: Feasibility and acceptability of using the BabySaver resuscitation platform and NeoBeat together for neonatal resuscitation in a low-resource setting: A pre-post implementation study

Author: Ritah Nantale

Summary: In this article the author(s) examined the feasibility of introducing the bedside neonatal resuscitation practice using the portable BabySaver resuscitation platform alongside the NeoBeat heart rate monitor, arguing that when combined, resulted in prompt initiation of neonatal resuscitation, a shorter duration to successful resuscitation with improved neonatal outcomes. The author(s) also explored the lived experiences of the healthcare workers performing the resuscitations and the mothers of respective infants and demonstrated the feasibility, ease of use and acceptability of this low cost intervention which fostered trust and satisfaction from the mothers. The article concludes that the use of BabySaver and NeoBeat intervention increased bedside resuscitation rates and significantly reduced neonatal morbidity .

Analysis: The article effectively illustrated that with pre-intervention training and incorporation of the new interventions into the SOPs, bedside resuscitation was easily adopted and implemented especially in a one-man resuscitation. It encompassed a much needed heart rate monitor to give ongoing feedback during the resuscitation, which was previously not readily available.The strength of this study lies in the quasi experimental approach where the authors demonstrated the outcomes of interest using the standard of care at baseline and compared the same outcomes post intervention. The uptake of bedside resuscitation was significantly higher which allowed the authors to demonstrate a relative improvement in the neonatal resuscitation process and which decreased neonatal morbidity and the need for admission. The buy-in by both healthcare workers and mothers is a cue for fostering sustainability

However, the article could benefit from some clarification on

1. How they arrived at a sample size estimation of 150 observations, and the entry point thereof.

2. How much power these 15 observations afforded to the study.

3. From the Figure 1, flow chart the entry point is the total neonatal resuscitations during each phase, of whom 150 consented in both the pre and post arms; and only 75 of each were included in the final analysis. So, how did the authors select the specific 75 observations in each arm out of the 150

Insights

This article illustrates the empowerment of healthcare workers in adopting bedside neonatal resuscitation using low cost interventions that significantly improve neonatal outcomes in LMICs. These findings were enhanced by a feedback loop from the qualitative context of the healthcare workers' and mothers' experiences. The study also dispelled some myths and misconceptions serving as potential barriers for the adoption of the intervention. Future research might explore a country-wide scale up of this intervention to determine its sustainability whilst incorporating suggested modifications.

Conclusion:

Overall the article provides a valuable contribution to the improvement of neonatal resuscitation outcomes using feasible and acceptable low cost intervention strategies in limited resource settings. Future research can explore the country-wide scale up of the intervention with incorporation of the suggested modifications which can be life saving.

This article meets the criteria to be considered for publication and will contribute to a gap in knowledge on neonatal resuscitation applicable in low resource settings. However, there is need for some minor corrections and clarifications as follows

Abstract : Is well written and concise

Background :

The narrative in the background did not allude to the fact that bedside resuscitation was already in practice pre- implementation as we find later in the results. Was this contributed by previous interventions or was this part of the standard of care?

It would be worthy to cite (Ditai J et al, 2021) who developed the BabySaver in the same hospital and inform the reader if its use had been adopted or not and what were the concerns and how have these been addressed in the current study? or if this is the first feasibility study on the BabySaver?

Grammar : Paragraph line 8 .... This practice prevents the practice of delyaed cord clamping............ The double use of practice in the same sentence can be improved

Problem statement and Objectives: Were well stated

Paragraph 4 in the Background section

Justification: This can be enhanced by citations that show the potential benefits of beside resuscitation in terms of morbidity indicators (early initiation, improved Apgars, decreased neonatal morbidity) and not just the delayed cord clamping benefits [ REF 13-15].

Methods

Study setting:

Line 5..... The hospital has one labour ward theatre ....... this phrase can be changed to an obstetric theatre a terminology which will resonate with most readers.

Sample size estimation and sampling needs clarification as mentioned in the above summary

Variables

Paragraph 2. ........A clarification between the baseline socio-demographic characteristics versus the independent and dependent variables is warranted to be able to interpret the tables correctly. Table 3 variables are dependent on whether a pre or post implementation approach was adopted and not the reverse.

Data collection: Was well elaborated

Data analysis For both phases was done well, using the appropriate/relevant statistical methods for the study design

Ethical consideration for intrapartum research was duly followed and clearance was obtained through the right pathways.

Results Overall the results were well presented

Study profile

Line 5 . in the post intervention arm only 152 mother-infant dyads were eligible not 155 and shown in Figure 1

Table 4 Comparison of the ...... birth outcomes pre and post implementation of the bedside neonatal resuscitation did not show any difference in the condition of the neonate at 24 and 48 hours. This was not discussed by the authors. Is there any explanation for this? Does it call for long-term follow up research ?

The qualitative findings were well presented and the use of an existing theoretical framework was ideal in streamlining the interpretations of the arising themes, although there was some overlap in the some constructs.

The suggested modifications added great value to the qualitative findings allowing the end user to contribute towards the quality improvement of the intervention

Discussion: Was brief but concise and focused on the responding to the main objective.

Typo ........

Paragraph 1 Line .5...... fewer babies needed further care post resuscitation

Paragraph 2 Line 16......Several studies including systematic reviews

Limitations: The use of the same researchers for the IDI may indeed have biased the participants. Is there a reason different research assistants were not allocated for this phase? If so, how was the potential for bias mitigated?

Conclusions and recommendations : We appropriate and derived from the study findings.

References: These were appropriately selected but it would help to know what reference style the authors used to assess for consistency?

Reviewer #2: This is a well written and important work. I believe the conclusions are sound and based on the research which is well described. I have a few important queries/observations, and then a number of less important minor issues.

I think the issues around the sample size need to be better clarified. What is the potential change in outcome that was used to calculate the sample size? Improved resuscitation at the bedside? And if so, where does the assumption of a 50% change come from? What alpha/power was used ?

I find the graph documenting differences in temperature very interesting. I am not clear whether the initial temperature is at the time of birth or at 1 minute. It would seem odd for there to be a difference in temperature at the time of birth. But – before intervention- it seems the initial temperature ( ? at 1 min) was 33 degrees? That seems noteworthy and I wonder if it is accurate?

It is reported that the duration of resuscitation was shorter post intervention. Were there specific objective criteria to determine when resuscitation stopped? Should these be listed?

I have a few questions around table 4. The footnotes all say “ adjusted for antepartum haemorrhage and malpresentation”. I suspect this is a typo. Perhaps the correct footnotes may answer some of my questions.

Under outcomes of “ the resuscitation process” it mentions early neonatal death. This seems less than the number of deaths at 24 h- does this line refer to babies who died in the resuscitation process? That should be made clear. And there were infact an increase in the number of deaths- albeit a small one from 0 to 3. But no p value given- I would think this is important, from my calculatiosn using a Fischers exact I get 0.244. I think this is worth mentioning in the text, even if it is not significant. I am also wondering why the total number of babies in the 24 h and 48h intervals does not add up to 75.

The rest of my comments are more minor and a matter of style.

In the methods section there seems to a semi- repetition of a sentence around the use of a phenomenological method.

There is mention of a resuscitation checklist in the methods- and then in the data it is mentioned that the liveborn LGH tool is used- if these are the same, I may mention the LGH tool in the methods.

In the table about characteristics of the mothers , some are classified as housewives and others as unemployed. What was the distinction used to separate these two categories?

I was often intrigued by what the products may look like- is there a possibility to show a photo of them, ideally in use? Or set up for use?

In the qualitative section, it is mentioned that 63% of the mothers had secondary education, but for the group as a whole I see it was more like 40%- should this be mentioned?

I also feel that the slightly different training for staff in the pre and post intervention period may have had some impact on the outcomes ( ie not just the gadgets themselves) and I think that might need to be mentioned.

**Do you want your identity to be public for this peer review?** For information about this choice, including consent withdrawal, please see our Privacy Policy

Reviewer #1: No

Reviewer #2: **Yes: ** Alexander G. Stevenson

---

## [Author Response · Author response to Decision Letter 1]

23 Jul 2025

Re: “PONE-D-25-16948 Feasibility and acceptability of using the BabySaver resuscitation platform and NeoBeat together for neonatal resuscitation in a low-resource setting: A pre-post implementation study”

Thank you for revising our manuscript.

Below is our point-by-point response to each comment.

Thank you.

Comment Response to comment Line number

Editor

1. Please correct the author affiliations: The author affiliations do not match - “9” and “11” are not assigned to an author. Thank you for pointing this out. We have reviewed and corrected the author affiliations. The numbers “9” and “11” have now been appropriately assigned to the relevant authors, and all affiliations are now correctly matched. 3-6

2. Please add further details to the funding, conflict of interest sections, and the text in the manuscript, to make appropriate reference to the website which describes the work of Andrew Weeks (the last author): https://safri.org/the-baby-saver/

The website states:

The BabySaver, developed by Professor Andrew Weeks, a consultant obstetrician from Liverpool.

The product has been developed in conjunction with Peter Watt, a design engineer at the Royal Liverpool and Broadgreen University Hospitals Trust, after trials with staff and patients at the Mbale Regional Referral Hospital (MRRH) in Uganda.

The prototypes have been manufactured by a team at the Bryn Y Neuadd Hospital in Gwynedd.

The project has been funded by Grand Challenges Canada, and the Sir Halley Stewart Trust.

NOTE:

i) Since Andrew Weeks is the designer/developer of the device which is being tested, that should be stated, as well as a clear statement regarding the agreements between him and the funders, including subsequent renumeration for sales of the device and any related patents.

ii) Since the website represents the Sanyu Africa Research Institute (SAfRI), the affiliations between this institute and the author(s) must be clarified. Thank you for this important comment, and apologies for overlooking this important issue.

We have clarified the affiliations with the Sanyu Africa Research Institute (SAfRI) and have updated the funding and conflict of interest sections at the end of the document. 647-663

Reviewer #1

1. How they arrived at a sample size estimation of 150 observations, and the entry point thereof.

How much power these 15 observations afforded to the study. Thank you for these comments regarding sample size estimation. The target sample size was 150 neonatal resuscitation observations. The sample size was based from local hospital data. Mbale Regional Referral Hospital conducts approximately 9,000 deliveries annually. Based on hospital estimates, about 6% of newborns (approximately 540) require resuscitation annually. Assuming that 50% of these would be eligible for bedside resuscitation using the BabySaver (n = 270), and considering a six-month data collection period, we aimed to observe around 135 resuscitation events. To account for potential missed observations, we added a 10% inflation, resulting in a final target sample size of 150 observations, with 75 in each phase (pre- and post-implementation). As a feasibility study, no formal sample size calculation was needed, but we considered that a sample of 75 in each group would be adequate to detect an increase in bedside resuscitation from 10% baseline to 30% after implementation (5% significance level and 90% power). 155-162

2. From the Figure 1, flow chart the entry point is the total neonatal resuscitations during each phase, of whom 150 consented in both the pre and post arms; and only 75 of each were included in the final analysis. So, how did the authors select the specific 75 observations in each arm out of the 150 Thank you for pointing this out. We sincerely acknowledge the concern and appreciate the opportunity to clarify this issue. The initial figure was a misrepresentation resulting from a miscommunication from the data management team. We have reviewed our data sources thoroughly to ensure that the corrected numbers accurately reflect actual enrollment. The actual number of participants who consented and were included in each phase was 75. During the pre-implementation phase, 81 mother–infant dyads were eligible, and 75 were enrolled. In the post-implementation phase, 78 mother–infant dyads were eligible, and 75 were enrolled. We have revised the manuscript and Figure 1 to accurately reflect the total number of enrolled participants (75 per phase). 255-263

3. Abstract: Is well written and concise Thank you for this feedback. We are glad the abstract is well written and concise. NA

4. The narrative in the background did not allude to the fact that bedside resuscitation was already in practice pre- implementation as we find later in the results. Was this contributed by previous interventions or was this part of the standard of care? Thank you for this important observation. We acknowledge that the background section did not explicitly state that bedside resuscitation practices, though limited, were already occurring prior to the implementation. The WHO recommends that delayed cord clamping should be practiced for every newborn and this has been emphasized through webinars prior to the intervention. However, bedside resuscitation for asphyxiated newborns was not routine and was constrained by the lack of appropriate equipment at the mother’s bedside. This study focused specifically on resuscitation of asphyxiated newborns, where the standard approach was still early cord clamping and transfer to a separate resuscitation area. We have included it in the study setting. 111-114

5. It would be worthy to cite (Ditai J et al, 2021) who developed the BabySaver in the same hospital and inform the reader if its use had been adopted or not and what were the concerns and how have these been addressed in the current study? or if this is the first feasibility study on the BabySaver? Thank you. We have cited Ditai J et al., 2021, and have clarified that this is the first feasibility study of BabySaver and NeoBeat together under the study strengths. Previous work focused on development and design. 85 and 572

6. Grammar: Paragraph line 8 .... This practice prevents the practice of delyaed cord clamping............ The double use of practice in the same sentence can be improved Thank you for highlighting this, we have revised it to; “This practice hinders delayed cord clamping and may also cause maternal anxiety and denial.” 81

7. Problem statement and Objectives: Were well stated Thank you for this feedback. We are happy that the problem statement and objectives are clear and well-stated. NA

8. Paragraph 4 in the Background section

Justification: This can be enhanced by citations that show the potential benefits of beside resuscitation in terms of morbidity indicators (early initiation, improved Apgars, decreased neonatal morbidity) and not just the delayed cord clamping benefits [REF 13-15]. We have revised and added citations demonstrating reduced neonatal morbidity (improved Apgar scores, and early initiation). 88-89

9. Line 5..... The hospital has one labour ward theatre ....... this phrase can be changed to an obstetric theatre a terminology which will resonate with most readers. Thank you for the suggestion, this has been done. 106

10. Sample size estimation and sampling needs clarification as mentioned in the above summary Thank you for your comment. We have revised the manuscript to provide a more detailed explanation of how the sample size of 150 observations (75 per phase) was estimated. The estimation was based on institutional data indicating approximately 9,000 annual deliveries at Mbale Regional Referral Hospital, with 6% requiring resuscitation (n = 540). Assuming 50% eligibility for bedside resuscitation, this gave an estimated 270 eligible cases over one year. Over the six-month study period, we expected around 135 eligible cases. We inflated this by 10% to account for missed or incomplete observations, resulting in a final target sample of 150 observations.

Eligible participants were consecutively enrolled until the required sample size of 75 per phase was achieved. As a feasibility study, no formal sample size calculation was needed, but we considered that a sample of 75 in each group would be adequate to detect an increase in bedside resuscitation from 10% baseline to 30% after implementation (5% significance level and 90% power). 155-162

11. Variables

Paragraph 2. ........A clarification between the baseline socio-demographic characteristics versus the independent and dependent variables is warranted to be able to interpret the tables correctly. Table 3 variables are dependent on whether a pre or post implementation approach was adopted and not the reverse. Thank you for this insightful comment. We agree that the distinction between baseline socio-demographic characteristics and the study’s independent and dependent variables needed clarification. We have revised Table 3 to show the study phase (pre or post) as the independent variable, and place of resuscitation as the dependent variable.

Additionally, we have clarified that the baseline socio-demographic characteristics presented in Table 1 serve a descriptive purpose to compare participant groups across study phases, rather than acting as predictors in the analysis.

We have included study phase as one of the explanatory variables. We have chosen to use the explanatory variables

290

174-176

12. Data collection: Was well elaborated Thank you, we appreciate your acknowledgment. NA

13. Data analysis For, both phases was done well, using the appropriate/relevant statistical methods for the study design Thank you, we are pleased that the data analysis approach was found appropriate and well done. NA

14. Ethical consideration for intrapartum research was duly followed and clearance was obtained through the right pathways. Thank you for acknowledging our efforts in adhering to ethical standards for intrapartum research. NA

15. Results Overall the results were well presented We appreciate your observation and are glad that the results are well presented. NA

16. Study profile

Line 5. in the post intervention arm only 152 mother-infant dyads were eligible not 155 and shown in Figure 1 Thank you for the observation. We have reviewed and corrected the study profile accordingly 264-265

17. Table 4 Comparison of the ...... birth outcomes pre and post implementation of the bedside neonatal resuscitation did not show any difference in the condition of the neonate at 24 and 48 hours. This was not discussed by the authors. Is there any explanation for this? Does it call for long-term follow up research? Thank you for this valuable comment. We acknowledge the omission and have now addressed this in the Discussion section. However, we did not observe differences in the condition of the infant at 24 and 48 hours. This may be due to the limitations of short-term outcome measures, which may not adequately reflect the benefits of timely bedside resuscitation. 517-519

18. The qualitative findings were well presented and the use of an existing theoretical framework was ideal in streamlining the interpretations of the arising themes, although there was some overlap in the some constructs. Thank you for your comment. NA

19. The suggested modifications added great value to the qualitative findings allowing the end user to contribute towards the quality improvement of the intervention Thank you for this feedback. NA

20. Discussion: Was brief but concise and focused on the responding to the main objective. Thank you for this comment. NA

21. Typo ........

Paragraph 1 Line .5...... fewer babies needed further care post resuscitation

Paragraph 2 Line 16......Several studies including systematic reviews We have revised these statements to: “Fewer babies required additional care post-resuscitation.” and, “Several studies, including systematic reviews…” as guided. 489 and 513

22. Limitations: The use of the same researchers for the IDI may indeed have biased the participants. Is there a reason different research assistants were not allocated for this phase? If so, how was the potential for bias mitigated? Yes, this was a possible source of bias However, we mitigated bias by training researchers on qualitative data collection techniques including the practice of reflexivity. We have updated this under discussion section. 584-586

23. Conclusions and recommendations: We appropriate and derived from the study findings. Thank you for your comment. NA

24. References: These were appropriately selected but it would help to know what reference style the authors used to assess for consistency? We used Vancouver referencing style throughout and have ensured consistency. NA

Reviewer #2

1. This is a well written and important work. I believe the conclusions are sound and based on the research which is well described. I have a few important queries/observations, and then a number of less important minor issues. We sincerely thank the reviewer for the positive and encouraging feedback on the overall quality, relevance, and clarity of our work. NA

2. I think the issues around the sample size need to be better clarified. What is the potential change in outcome that was used to calculate the sample size? Improved resuscitation at the bedside? And if so, where does the assumption of a 50% change come from? What alpha/power was used? Thank you for this important observation. We acknowledge that further clarification was needed regarding the assumptions underlying the sample size calculation. We have revised it and now reads; Based on the hospital’s annual number deliveries (9,000), the percentage of neonates (6%) that require resuscitation in Mbale Regional Referral Hospital (local data) (9000*0.06=540). Given the assumption that 50% of eligible neonates would receive bedside resuscitation (540*0.5=270). Over a period of six months, we estimated to observe (270/2=135). We inflated the sample size by 10 percent to cater for missed observations. The required sample size was 150 observations, with 75 in each phase. We assumed a two-sided confidence interval of 95%, and 99.9% power using a before and after approach. We consecutively enrolled the mother-infant dyads until the required sample size was accrued in each phase. As a feasibility study, no formal sample size calculation was needed, but we considered that a sample of 75 in each group would be adequate to detect an increase in bedside resuscitation from 10% baseline to 30% after implementation (5% significance level and 90% power). 156-164

3. I find the graph documenting differences in temperature very interesting. I am not clear whether the initial temperature is at the time of birth or at 1 minute. It would seem odd for there to be a difference in temperature at the time of birth. But – before intervention- it seems the initial temperature (? at 1 min) was 33 degrees? That seems noteworthy and I wonder if it is accurate? Thank you. Initial temperature was recorded within the golden minute. This observation of low temperature may reflect heat loss during transfer to the resuscitation after early cord clamping in the pre-implementation phase. However, in the discussion, we postulate that babies resuscitated with an intact cord receive a transfusion of warm blood form the mother and may have a better regulation of their temperature. 520-527

4. It is reported that the duration of resuscitation was shorter post intervention. Were there specific objective criteria to determine when resuscitation stopped? Should these be listed? We were able to tell the duration of resuscitation more accurately using the Liveborn app, that we used to capture time stamps in both the pre and post implementation phase. 187-189

Reviewer #3

1. I have a few questions around table 4. The footnotes all say “adjusted for antepartum haemorrhage and malpresentation”. I suspect this is a typo. Perhaps the correct footnotes may answer some of my questions. We apologize for this oversight. This has been clarified under the statistical analysis section, antepartum haemorrhage and malpresentation were adjusted for, in all the models. 224-225

2. Under outcomes of “the resuscitation process” it mentions early

---

## [Decision Letter · Decision Letter 1]

8 Aug 2025

Dear Dr. Nantale,

Thank you for submitting your manuscript to PLOS ONE. After careful consideration, we feel that it has merit but does not fully meet PLOS ONE’s publication criteria as it currently stands. Therefore, we invite you to submit a revised version of the manuscript that addresses the points raised during the review process.

**Please respond to the remaining issues raised by the reviewers, including the point raised by reviewer one, as a proviso to their acceptance.**
**Please also update the response to conflict of interest section of the submission, which should be "Yes" there is  a potential **
**conflict which must be stated as: **

"ADW holds no personal rights to the IP, but there is a royalty sharing scheme at UoL whereby any future royalties would be shared with him." In addition the amount and/or terms of those royalties needs to be transparently stated.

We look forward to receiving your revised manuscript.

Kind regards,

Alan Richard Horn, MD, DCH, FCPaed, Cert. Neonatology, PhD

Academic Editor

PLOS ONE

Journal Requirements:

Reviewers' comments:

Reviewer's Responses to Questions

**Comments to the Author**

Reviewer #1: All comments have been addressed

Reviewer #2: (No Response)

2. Is the manuscript technically sound, and do the data support the conclusions?

Reviewer #1: Yes

Reviewer #2: Yes

3. Has the statistical analysis been performed appropriately and rigorously?

Reviewer #1: Yes

Reviewer #2: Yes

4. Have the authors made all data underlying the findings in their manuscript fully available?

Reviewer #1: Yes

Reviewer #2: Yes

5. Is the manuscript presented in an intelligible fashion and written in standard English?

Reviewer #1: Yes

Reviewer #2: Yes

Reviewer #1: Thank you , the author(s) have addressed the comments stipulated and they have cleared out the sample size and sampling queries which were previously confusing. Similarly the revision of Table 4 was a crucial step in clarifying the key outcomes,

1, one minor comment in the background line 81 .................. This practice hinders delayed cord-clamping and may also cause mother anxiety and denial CHANGE TO..........to maternal anxiety

2. In the results , Line 311-312, the narrative on the lack of a difference in outcome at 24 hours and 48 hours should still be documented. This has been rightfully included in the discussion, though it is my opinion that HIE is an evolving condition and the outcome at 24 and 48 hours may be influenced by affection of other organ systems and the management thereof; and is not limited to the initial resuscitation results.

In conclusion

The authors have produced a good manuscript with important scientific evidence and have addressed the major concerning issues

Reviewer #2: Thank you for addressing my previous comments.

I still feel the claim that baby's axillary temperature at 1 min was 33.5 degrees warrants a comment in the discussion. I strongly suspect this was a measuring error. Even babies who are actively cooled with icepacks take 30 min to drop their temperature that far. I find it hard to accept that a baby no more than 1 min old, who presumably had a normal temperature 1 min ago, could drop their temperature to 33 degrees in 1 minute.

**Do you want your identity to be public for this peer review?** For information about this choice, including consent withdrawal, please see our Privacy Policy

Reviewer #1: No

Reviewer #2: **Yes: ** alexander g stevenson

---

## [Author Response · Author response to Decision Letter 2]

3 Sep 2025

Re: “PONE-D-25-16948 Feasibility and acceptability of using the BabySaver resuscitation platform and NeoBeat together for neonatal resuscitation in a low-resource setting: A pre-post implementation study”

Thank you for revising our manuscript.

Below is our point-by-point response to each comment.

Thank you.

Comment Response to comment Line number

Reviewer #1

1. Thank you, the author(s) have addressed the comments stipulated and they have cleared out the sample size and sampling queries which were previously confusing. Similarly, the revision of Table 4 was a crucial step in clarifying the key outcomes,

1, one minor comment in the background line 81 .................. This practice hinders delayed cord-clamping and may also cause mother anxiety and denial CHANGE TO..........to maternal anxiety Thank you for this feedback.

Thank you for the suggestion, we have revised this sentence and it now reads; “This practice hinders delayed cord-clamping and may also cause maternal anxiety and denial.”

81

2. In the results, Line 311-312, the narrative on the lack of a difference in outcome at 24 hours and 48 hours should still be documented. This has been rightfully included in the discussion, though it is my opinion that HIE is an evolving condition and the outcome at 24 and 48 hours may be influenced by affection of other organ systems and the management thereof; and is not limited to the initial resuscitation results. Thank you for pointing this out. We have added a statement under results highlighting that: “There was no difference in the condition of the infant at 24 and 48 hours between the pre- and post- implementation periods.” 314-315

Reviewer #2

1. Thank you for addressing my previous comments.

I still feel the claim that baby's axillary temperature at 1 min was 33.5 degrees warrants a comment in the discussion. I strongly suspect this was a measuring error. Even babies who are actively cooled with icepacks take 30 min to drop their temperature that far. I find it hard to accept that a baby no more than 1 min old, who presumably had a normal temperature 1 min ago, could drop their temperature to 33 degrees in 1 minute. Thank you for this comment.

We suspect hat this was due to the fluid on the babies’ foreheads immediately after birth. We have included this as a limitation and state it as below:

“One limitation of infra-red temperature assessment is the tendency to underestimate readings in the presence of moisture. We took our first temperature measurement immediately after birth, before drying the baby. This could have resulted in a systematic underestimation of the temperature reading at 1 minute. However, this error would have been similar in both the pre-implementation and post-implementation period and we suspect that this did not affect the difference between the two periods.” 260-265

---

## [Decision Letter · Decision Letter 2]

1 Oct 2025

Dear Dr. Nantale,

Thank you for submitting your manuscript to PLOS ONE. After careful consideration, we feel that it has merit but does not fully meet PLOS ONE’s publication criteria as it currently stands. Therefore, we invite you to submit a revised version of the manuscript that addresses the points raised during the review process.

**Please address the following remaining aspects in your final revision:**

We look forward to receiving your revised manuscript.

Kind regards,

Alan Richard Horn, MD, DCH, FCPaed, Cert. Neonatology, PhD

Academic Editor

PLOS ONE

**Journal Requirements:**

Reviewers' comments:

Reviewer's Responses to Questions

**Comments to the Author**

Reviewer #1: All comments have been addressed

Reviewer #2: All comments have been addressed

2. Is the manuscript technically sound, and do the data support the conclusions?

Reviewer #1: Yes

Reviewer #2: Yes

3. Has the statistical analysis been performed appropriately and rigorously?

Reviewer #1: Yes

Reviewer #2: Yes

4. Have the authors made all data underlying the findings in their manuscript fully available?

Reviewer #1: Yes

Reviewer #2: Yes

5. Is the manuscript presented in an intelligible fashion and written in standard English?

Reviewer #1: Yes

Reviewer #2: Yes

**Reviewer #1:**  Greetings

Thank you for working on the previous comments which have added great clarity to your work thus far.

I have very minor comments which you can consider for your final version.

Few corrections

Abstract

Line 45: the median time to successful resuscitation was shorter 8 versus 5 minutes, [I think should be the reverse '5 versus 8 minutes]'

Background

Line 66: hindered by a lack of training and suitable and functional equipment [ 'and suitable or functional resuscitation equipment '] may be used to avoid the double and in the sentence

Study population

Line 116 admitted to the neonatal unit ['for'] assessment and ongoing management... the word for is missing

Study procedures

Lines 122 and 136 ['obstetric theatre'] rather than labour ward theatre

One inquiry

Background

Line 85 :Just wanted clarity why it is said the Babysaver has not been clinically tested whilst it had undergone phase 1 and 11 clinical testing and part of this was done in Uganda?

Otherwise, my previous comments have been addressed

Good luck

**Reviewer #2: ** all issues have been addressed, there are no further comments and the submission is ready for publication as it is.

**Do you want your identity to be public for this peer review?** For information about this choice, including consent withdrawal, please see our Privacy Policy

Reviewer #1: No

Reviewer #2: **Yes: ** alexander g stevenson

---

## [Author Response · Author response to Decision Letter 3]

6 Oct 2025

Re: “PONE-D-25-16948 Feasibility and acceptability of using the BabySaver resuscitation platform and NeoBeat together for neonatal resuscitation in a low-resource setting: A pre-post implementation study”

Thank you for revising our manuscript.

Below is our point-by-point response to each comment.

Thank you.

Comment Response to comment Line number

Editor

Correct the conflict of interest statement in the standardised table before the manuscript so that it aligns with the conflict of interest statement at the end of the manuscript (the receipt of royalties is a potential conflict of interest).

Thank you for your comment, this has been done.

Reviewer #2

Thank you for working on the previous comments which have added great clarity to your work thus far.

I have very minor comments which you can consider for your final version

Few corrections

Abstract

Line 45: the median time to successful resuscitation was shorter 8 versus 5 minutes, [I think should be the reverse '5 versus 8 minutes]'

1.

Background

Line 66: hindered by a lack of training and suitable and functional equipment [ 'and suitable or functional resuscitation equipment '] may be used to avoid the double and in the sentence

Study population

Line 116 admitted to the neonatal unit ['for'] assessment and ongoing management... the word for is missing

Study procedures

Lines 122 and 136 ['obstetric theatre'] rather than labour ward theatre

One inquiry

Background

Line 85:Just wanted clarity why it is said the BabySaver has not been clinically tested whilst it had undergone phase 1 and 11 clinical testing and part of this was done in Uganda? Thank you for this feedback

Thank you, these grammatical errors have been revised accordingly.

45

68

119

125, 136

Thank you for the comment. There had been no clinical testing conducted prior to this study. However, the design and development of the BabySaver was partly done out in Uganda.

260-265

---

## [Editor Report · Decision Letter 3]

12 Oct 2025

Dear Dr. Nantale,

Thank you for submitting your manuscript to PLOS ONE. After careful consideration, we feel that it has merit but does not fully meet PLOS ONE’s publication criteria as it currently stands. Therefore, we invite you to submit a revised version of the manuscript that addresses the points raised during the review process.

**Please correct the competing interests section on page 5 of  PONE-D-25-16948R3 still reads, "The authors have declared that no competing interests exist." Please correct this to align with the statement in the manuscript which reads, **

**"ADW was one of the co-inventors of the BabySaver tray. The intellectual property is held by**

**his employer, the University of Liverpool, but the rights for Africa were sold to the Sanyu**

**Africa Research Institute for £1 in 2019 so that they could take forward its development and**

**distribution in Africa. ADW holds no personal rights to the IP, but there is a royalty sharing**

**scheme at UoL whereby any future royalties would be shared with him."**

We look forward to receiving your revised manuscript.

Kind regards,

Alan Richard Horn, MD, DCH, FCPaed, Cert. Neonatology, PhD

Academic Editor

PLOS ONE
---

## [Author Response · Author response to Decision Letter 4]

17 Oct 2025

Re: “PONE-D-25-16948 Feasibility and acceptability of using the BabySaver resuscitation platform and NeoBeat together for neonatal resuscitation in a low-resource setting: A pre-post implementation study”

Thank you for revising our manuscript.

Below is our point-by-point response to each comment.

Thank you.

Comment Response to comment Line number

Editor

Please correct the competing interests’ section on page 5 of PONE-D-25-16948R3 still reads, "The authors have declared that no competing interests exist." Please correct this to align with the statement in the manuscript which reads,

"ADW was one of the co-inventors of the BabySaver tray. The intellectual property is held by his employer, the University of Liverpool, but the rights for Africa were sold to the Sanyu Africa Research Institute for £1 in 2019 so that they could take forward its development and

distribution in Africa. ADW holds no personal rights to the IP, but there is a royalty sharing scheme at UoL whereby any future royalties would be shared with him." Thank you for your comment. We are unable to locate where to update the Competing Interests section in the submission portal. We previously contacted PLOS Customer Care and were advised to include the updated statement in the cover letter, after which the PLOS team would make the update on our behalf.

Their response was as follows:

“Thank you for your query. The Competing Interests statement can't be edited on your end past initial submission, so please include your new statement in your cover letter and we will update this for you.”

We added the updated statement to the cover letter.

We kindly seek your guidance on how best to proceed in this regard.

---

## [Editor Report · Decision Letter 4]

4 Nov 2025

Feasibility and acceptability of using the BabySaver resuscitation platform and NeoBeat together for neonatal resuscitation in a low-resource setting: A pre-post implementation study

PONE-D-25-16948R4

Dear Dr. Nantale,

We’re pleased to inform you that your manuscript has been judged scientifically suitable for publication and will be formally accepted for publication once it meets all outstanding technical requirements.

Kind regards,

Alan Richard Horn, MD, DCH, FCPaed, Cert. Neonatology, PhD

Academic Editor

PLOS ONE
---

## [Editor Report · Acceptance letter]

PONE-D-25-16948R4

PLOS ONE

Dear Dr. Nantale,

I'm pleased to inform you that your manuscript has been deemed suitable for publication in PLOS ONE. Congratulations! Your manuscript is now being handed over to our production team.

Kind regards,

on behalf of

Professor Alan Richard Horn

Academic Editor

PLOS ONE